# Shared input and recurrency in neural networks for metabolically efficient information transmission

**Tomas Barta**[1,2]*, **Lubomir Kostal**[1]*

**1** Laboratory of Computational Neuroscience, Institute of Physiology of the Czech Academy of Sciences, Prague, Czech Republic, **2** Neural Coding and Brain Computing Unit, Okinawa Institute of Science and Technology, Onna-son, Okinawa, Japan

* tomas.barta@oist.jp (TB); kostal@biomed.cas.cz (LK)

## Abstract

Shared input to a population of neurons induces noise correlations, which can decrease the information carried by a population activity. Inhibitory feedback in recurrent neural networks can reduce the noise correlations and thus increase the information carried by the population activity. However, the activity of inhibitory neurons is costly. This inhibitory feedback decreases the gain of the population. Thus, depolarization of its neurons requires stronger excitatory synaptic input, which is associated with higher ATP consumption. Given that the goal of neural populations is to transmit as much information as possible at minimal metabolic costs, it is unclear whether the increased information transmission reliability provided by inhibitory feedback compensates for the additional costs. We analyze this problem in a network of leaky integrate-and-fire neurons receiving correlated input. By maximizing mutual information with metabolic cost constraints, we show that there is an optimal strength of recurrent connections in the network, which maximizes the value of mutual information-per-cost. For higher values of input correlation, the mutual information-per-cost is higher for recurrent networks with inhibitory feedback compared to feedforward networks without any inhibitory neurons. Our results, therefore, show that the optimal synaptic strength of a recurrent network can be inferred from metabolically efficient coding arguments and that decorrelation of the input by inhibitory feedback compensates for the associated increased metabolic costs.

## Author summary

Information processing in neurons is mediated by electrical activity through ionic currents. To reach homeostasis, neurons must actively work to reverse these ionic currents. This process consumes energy in the form of ATP. Typically the more energy the neuron can use, the more information it can transmit. It is generally assumed that due to evolutionary pressures, neurons evolved to process and transmit information efficiently at high rates but also at low costs. Many studies have addressed this balance between transmitted information and metabolic costs for the activity of single neurons. However, information

**Data Availability Statement:** The code is available at https://zenodo.org/records/10646689.

**Funding:** This work was supported by Charles University, project GA UK No. 1042120 granted to

TB. This article is published with financial support from the Strategy AV 21 Programme, "Breakthrough Technologies for the Future – Sensing, Digitisation, Artificial Intelligence, and Quantum Technologies". The funders had no role in study design, data collection and analysis, decision to publish, or preparation of the manuscript.

**Competing interests:** The authors have declared that no competing interests exist.

is often carried by the activity of a population of neurons instead of single neurons, and few studies investigated this balance in the context of recurrent neural networks, which can be found in the cortex. In such networks, the external input from thalamocortical synapses introduces pairwise correlations between the neurons, complicating the information transmission. These correlations can be reduced by inhibitory feedback through recurrent connections between inhibitory and excitatory neurons in the network. However, such activity increases the metabolic cost of the activity of the network. By analyzing the balance between decorrelation through inhibitory feedback and correlation through shared input from the thalamus, we find that both the shared input and inhibitory feedback can help increase the information-metabolic efficiency of the system.

# 1 Introduction

The efficient coding hypothesis poses that neurons evolved due to evolutionary pressure to transmit information as efficiently as possible [1]. Moreover, the brain has only a limited energy budget, and neural activity is costly [2, 3]. The metabolic expense associated with neural activity should, therefore, be considered, and neural systems likely work in an information-metabolically efficient manner, balancing the trade-off between transmitted information and the cost of the neural activity [4, 5, 6, 7, 8].

The principles of information-metabolically efficient coding have been successfully applied to study the importance of the excitation-inhibition balance in neural systems. It has been shown that the mutual information between input and output per unit of cost for a single neuron is higher if the excitatory and inhibitory synaptic currents to the neuron are approximately equal if the source of noise lies in the stochastic nature of the voltage-gated $Na^+$ and $K^+$ channels [9]. In a rate coding scheme, where the source of noise lies in the random arrival of pre-synaptic action potentials, the mutual information per unit of cost has been shown to be rather unaffected by the increase of pre-synaptic inhibition associated with an excitatory input [10].

However, the balance of excitation and inhibition is likely to be more important in the context of recurrent neural networks than in the context of single neurons. In recurrent neural networks, the inhibitory input to neurons associated with a stimulus [11] arises as inhibitory feedback from a population of inhibitory neurons. The inhibitory feedback prevents a self-induced synchronization of the neural activity [12] and reduces noise correlations (correlations between neurons calculated across trials of the same stimulus) induced by shared input to neurons in the population [13, 14, 15]. If noise correlations have the same sign as signal correlations (correlations between neurons calculated across different stimuli), then noise correlations are detrimental to information transmission by neural populations [16, 17, 18]. Information is likely transmitted by the activity of a population of neurons instead of a single neuron [19], therefore, when studying the effect of excitation-inhibition balance on information transmission, it is essential to consider the context of neural populations. In the case of a population of neurons tuned to the same stimulus, positive noise correlations decrease the information content in the population.

Several studies have analyzed the effect of noise correlations on information transmission properties [16, 17, 20]. However, these studies did not analyze the relationship between the noise correlations and the metabolic cost of neural activity. In our work, we consider a computational model of a small part of the sensory cortex and the noise correlations caused by shared connections from an external thalamic population. The noise correlations may then be reduced by inhibitory feedback, which, however, increases the cost of the neural activity [10].

Our point of interest is the trade-off between improved information transmission due to lower noise correlations and the increase in metabolic costs due to stronger inhibitory feedback.

## 2 Results

### 2.1 Constrained information maximization in a simple linear model

In order to gain an insight into what affects the information-metabolic efficiency of a neural population, we first solve the problem for a simple linear system. The mean response of the system is given by $\gamma(\lambda_{\text{ext}}) = g\lambda_{\text{ext}}$, where $\lambda_{\text{ext}}$ is the stimulus and $g$ is the gain of the system. We measure the trial-to-trial variability of the response with the Fano factor, defined as

$$\text{FF} = \frac{\text{Var}[N]}{\text{E}[N]}, \tag{1}$$

where $N$ is a random variable representing the response $n$ of the network to some stimulus. In this section, we assume the Fano factor to be constant, and we assume that the output is continuous and normally distributed. Therefore, the input-output relationship is described by the conditional probability

$$f(n|\lambda_{\text{ext}}) = \frac{1}{\sqrt{2g\lambda_{\text{ext}}\text{FF}}} \exp\left[-\frac{1}{2}\left(\frac{n - g\lambda_{\text{ext}}}{g\lambda_{\text{ext}}\text{FF}}\right)^2\right]. \tag{2}$$

We assume that the cost of the activity $w(\lambda_{\text{ext}})$ depends linearly on the input:

$$w(\lambda_{\text{ext}}) = w_0\lambda_{\text{ext}} + W_0 = \frac{w_0}{g}\gamma(\lambda_{\text{ext}}) + W_0, \tag{3}$$

where $W_0$ is the cost of the resting state.

We treat the input $\lambda_{\text{ext}}$ as a random variable $\Lambda$ with probability distribution function $p(\lambda_{\text{ext}})$. We can then calculate the average metabolic cost as

$$W_p = \int_{\lambda_{\text{ext}}^{\min}}^{\lambda_{\text{ext}}^{\max}} p(\lambda_{\text{ext}})w(\lambda_{\text{ext}})\,\mathrm{d}\lambda_{\text{ext}}. \tag{4}$$

The mutual information between the input and the output $I(\Lambda; N)$ is calculated as

$$I(\Lambda; N) = \int_{\lambda_{\text{ext}}^{\min}}^{\lambda_{\text{ext}}^{\max}} p(\lambda_{\text{ext}})i(\lambda_{\text{ext}}; N)\,\mathrm{d}\lambda_{\text{ext}}, \tag{5}$$

$$i(\lambda_{\text{ext}}; N) = \sum_{n=0}^{+\infty} i(\lambda_{\text{ext}}; n)q_p(n), \tag{6}$$

$$i(\lambda_{\text{ext}}; n) = \log_2 \frac{f(n|\lambda_{\text{ext}})}{q_p(n)}, \tag{7}$$

$$q_p(n) = \int_{\lambda_{\text{ext}}^{\min}}^{\lambda_{\text{ext}}^{\max}} p(\lambda_{\text{ext}})f(n|\lambda_{\text{ext}})\,\mathrm{d}\lambda_{\text{ext}}, \tag{8}$$

where $f(n|\lambda_{\text{ext}})$ is the probability distribution function of $N$ given that $\Lambda = \lambda_{\text{ext}}$, $p(\lambda_{\text{ext}})$ is the input probability distribution, $i(\lambda_{\text{ext}}; n)$ is the amount of information that an observation of $n$

spikes gives us about the stimulus $\lambda_{\text{ext}}$, $i(\lambda_{\text{ext}}; N)$ is then the average amount of information we get from the input $\lambda_{\text{ext}}$, $q_p(n)$ is the marginal output probability distribution.

The capacity-cost function $C(W)$ is the lowest upper bound on the amount of mutual information (in bits) achievable given the constraint that $W_p < W$:

$$C(W) = \sup_{p(\lambda_{\text{ext}}): W_p < W} I(\Lambda; N).$$ (9)

The information-metabolic efficiency $E$ is then the maximal amount of mutual information per molecule of ATP between the input and the output:

$$E = \frac{C(W^*)}{W^*},$$ (10)

$$W^* = \arg\max_{W \in [0, +\infty)} \frac{C(W)}{W}.$$ (11)

The capacity-cost function can be obtained numerically with the Blahut-Arimoto algorithm [21]. The information-metabolic efficiency can be conveniently obtained directly with the Jimbo-Kunisawa algorithm [22, 23]. However, if the Fano factor is very small, a lower bound on the capacity-cost function can be found analytically [24, 25]. In the low noise approximation, the optimal input distribution maximizing the mutual information constrained by metabolic expenses $W$ is given by

$$p(\lambda_{\text{ext}}) = \sqrt{\frac{J(\lambda_{\text{ext}})}{2\pi e}} \exp\left[\lambda_1 - 1 - \lambda_W w(\lambda_{\text{ext}})\right].$$ (12)

where $J(\lambda_{\text{ext}})$ is the Fisher information and $\lambda_1$ and $\lambda_W$ are the Lagrange multipliers which can be obtained from the normalization condition:

$$\int_{\lambda_{\text{ext}}^{\min}}^{\lambda_{\text{ext}}^{\max}} p(\lambda_{\text{ext}}) w(\lambda_{\text{ext}}) \, d\lambda_{\text{ext}}$$ (13)

and the average metabolic cost constraint (Eq 4). In the second-moment approximation [26, 27], the Fisher information is given by

$$J(\lambda_{\text{ext}}) = \frac{\mu'(\lambda_{\text{ext}})^2}{\sigma_{\text{exc}}(\lambda_{\text{ext}})^2},$$ (14)

where $\mu(\lambda_{\text{ext}})$ is the mean response to the external input $\lambda_{\text{ext}}$, $\mu'(\lambda_{\text{ext}})$ is the derivative, and $\sigma_{\text{exc}}(\lambda_{\text{ext}})$ is the standard deviation of the spike counts at input intensity $\lambda_{\text{ext}}$. The low noise estimate on the capacity-cost function is then

$$C_{\text{low}}(W) = 1 - \lambda_1 + \lambda_W W.$$ (15)

the information-metabolic efficiency can be conveniently obtained directly with the Jimbo-Kunisawa algorithm [22, 23].

In the case of our simple linear system the Fisher information (Eq 14) is

$$J(\lambda_{\text{ext}}) = \frac{g}{\lambda_{\text{ext}} \text{FF}},$$ (16)

and the probability distribution derived from the low-noise approximation (Eq 12) is then

$$p(\lambda_{\text{ext}}) = \sqrt{\frac{1}{2\pi e} \frac{g}{\lambda_{\text{ext}} \text{FF}}} \exp\left(\lambda_1 - 1 - \lambda_W w_0 \lambda_{\text{ext}}\right) \exp(-\lambda_W W_0). \tag{17}$$

After applying the normalization conditions (Eqs 4 and 13) and using Eq (15) we obtain the lower bound on the capacity-cost function:

$$C_{\text{low}}(W) = \frac{1}{2} \log \left[ (W - W_0) \frac{1}{w_{\text{AP}}} \frac{1}{\text{FF}} \right], \tag{18}$$

$$w_{\text{AP}} = \frac{w_0}{g}, \tag{19}$$

where $w_{\text{AP}}$ is the cost of increasing the output intensity by one action potential.

The gain $g$, cost scaling $w_0$, and Fano factor FF cannot be considered constant for real neural populations. However, Eq (18) provides an insight into the importance of these properties, which we will study numerically for a more realistic neural system.

In the following, we use

$$g = \mu'_{\text{ext}}(\lambda_{\text{ext}}), \tag{20}$$

$$w_0 = w'(\lambda_{\text{ext}}). \tag{21}$$

Next, we analyze the information-metabolic efficiency of a recurrent spiking neural network, consisting of 800 excitatory and 200 inhibitory neurons. This network may represent a small area in the cortex, tuned to the same external stimulus, such as approximately a sphere of a 145 μm radius in the rat barrel cortex, which comprises only a small fraction of a single barrel [28, 29]. In such case, the external input is the input from a single barreloid in the thalamus. We assume that the role of this subnetwork is to process information about the stimulus intensity. We analyze the information-metabolic efficiency in two extreme cases of the readout of the network. First, we assume that the output of the network is read out as the summed rate of all the neurons in the network, and second, we assume that the brain acts as an efficient unbiased decoder with access to the rate of each neuron. In each case, we calculate the rate of each neuron as the number of fired spikes in a time window $\Delta T = 1$ s.

## 2.2 Inhibitory feedback decorrelates the neural activity

In our model, 1000 external neurons randomly connect to the excitatory and inhibitory subpopulations with a connection probability $P_{\text{ext}}$ (Fig 1). Increasing $P_{\text{ext}}$ increases the mean pairwise correlation between the rates of the neurons in the network (feedforward network, Fig 1B). These correlations could be removed by recurrent connections. Initially, we set the excitatory recurrent synaptic amplitude as $a_{\text{exc}} = 0.01$ nS to create a small perturbation from the feedforward network and varied the scaling $\alpha$ determining the amplitude of inhibitory synapses ($a_{\text{inh}} = \alpha a_{\text{exc}}$) from 15 to 25, which leads to the amplitude of inhibitory post-synaptic potentials being sever-fold (approximately 2× to 8×, depending on $\alpha$ and on the memory potential) larger than the excitatory post-synaptic potentials, as commonly chosen in network modelling [30, 31, 32, 29]. Correlations between neurons were decreased for $\alpha \geq 20$ (Fig 1C), which was also associated with stronger negative net current from the recurrent synapses (Fig 1D). For the network considered further in our work we set $\alpha = 20$. Simultaneously increasing the strength of the recurrent synapses with fixed $\alpha$ led to a further decrease of the correlations

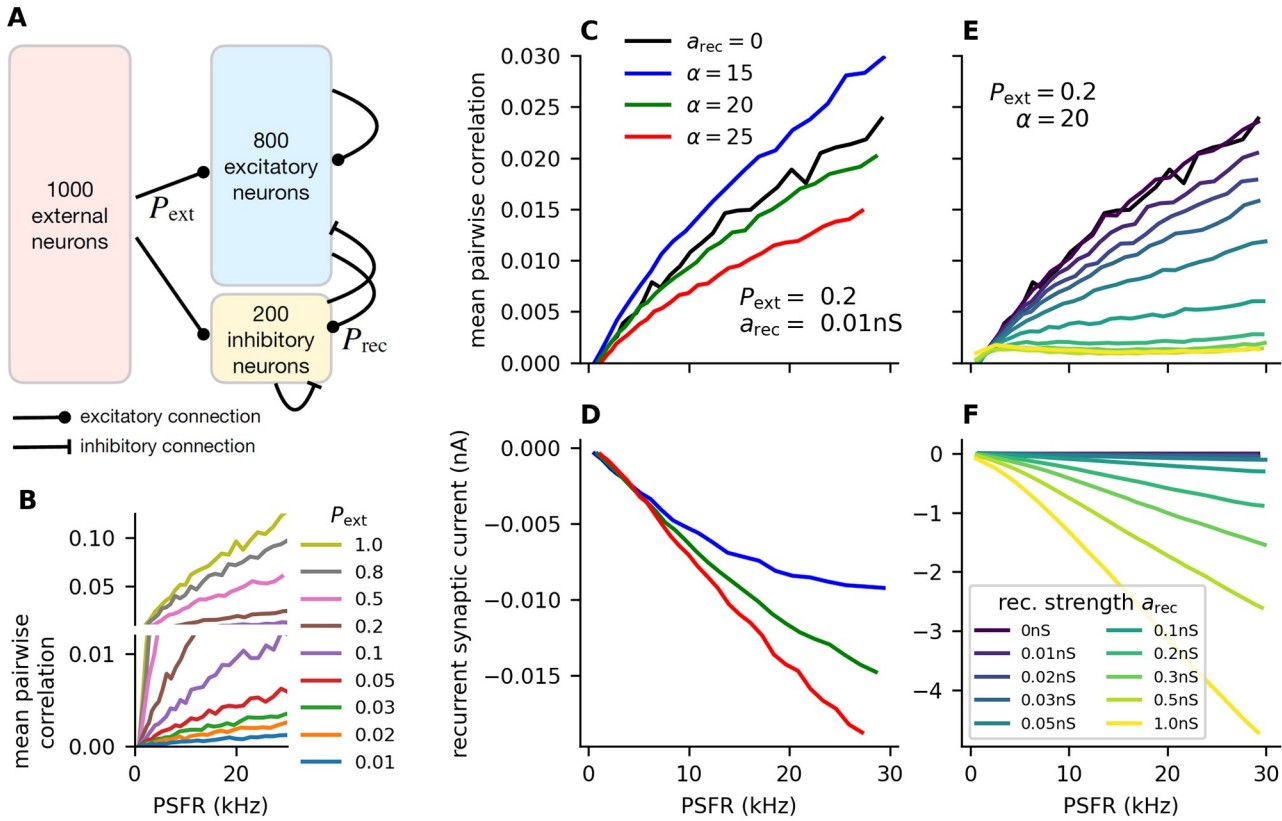

**Fig 1. Inhibitory feedback decreases noise correlations. A**: Schematic illustration of the simulated neural network. Poisson neurons in the external population make random connections to neurons in the excitatory and inhibitory subpopulations. The connection probability $P_{ext} \in [0.01, 1]$ is varied to achieve different levels of shared external input to the neurons. The neurons in the inhibitory (inh.) and excitatory (exc.) subpopulations make recurrent connections (exc. to exc., exc. to inh., inh. to inh., inh. to exc.) with probability $P_{rec} = 0.2$. The strength of those connections is parametrized by $a_{rec}$. **B**: Mean pairwise correlations between any two neurons in the exc. and inh. subpopulations plotted against the mean output of the network for different values of $P_{ext}$ in a feedforward network ($a_{rec} = 0$ nS). Pairwise correlations are calculated from the number of spikes each neuron fires in a time window $\Delta T = 1$ s across many trials of the simulation. The plot is vertically separated into two parts to also illustrate the smaller differences at lower values of $P_{ext}$. **C**: Mean pairwise correlations as in **B**, for different values of $\alpha$ (ratio of inhibitory-to-excitatory synaptic strength), $a_{rec} = 0.01$ nS. The black line represents the pairwise correlations in a feedforward network without any recurrent connections ($a_{rec} = 0$). **D**: Total current from recurrent synapses for different values of $\alpha$, as in **C**. **E-F**: Same as in **C-D**, but with fixed $\alpha = 20$ and different values of $a_{rec}$.

among the neurons (Fig 1E) while further decreasing the net current from the recurrent synapses (Fig 1F).

## 2.3 Fano factor of single neurons vs. a population

In an inhibition-dominated network, the input needed from the external population in order to evoke a given average firing rate has to be higher than in the case of the feedforward network. The resulting increase in synaptic noise leads to higher Fano factor in the LIF model (Fig 2A, 2B and 2C; see also [33]).

If we assume that the downstream areas decode the stimulus intensity from the summed activity of the network, we need to look at the Fano factor of the summed activity, that is, ratio of variance of the sum to the mean of the sum across the trials of duration $\Delta T = 1$ s. In the case of the total population activity, however, the pairwise correlations between the neurons have a significant effect on the Fano factor. By denoting the random variable representing the number of spikes of the $i$-th neuron observed during time window $\Delta T$ as $N_i$, we get for the Fano factor

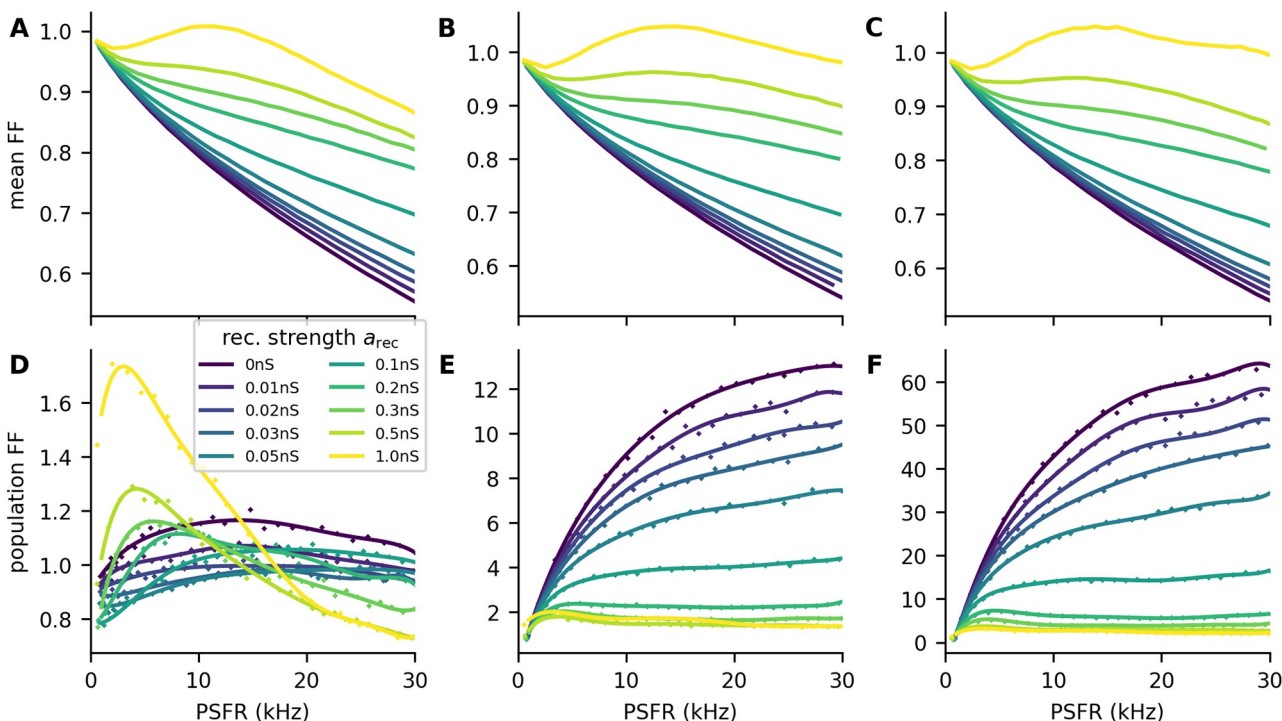

**Fig 2. Fano factor of single neurons and of populations. A-C**: Mean Fano factor of individual neurons for different values of $P_{ext}$: 0.01 (**A**), 0.2 (**B**), 1 (**C**). The strength of the recurrent synapses ($a_{rec}$) is color-coded. The mean Fano factor increases with the strength of the recurrent synapses. **D-F**: Same as in **A-C** but for the Fano factor of the population activity. The points represent the population Fano factor obtained from the simulation, and the lines are a weighted 7th-degree polynomial, used only as a visual aid. For $P_{ext} = 0.01$, the increase in Fano factor of individual neurons (**A**) can have a stronger effect on the population Fano factor than decreasing the pairwise correlations, resulting in an increase of the population Fano factor with high values of $a_{rec}$ (**D**). For higher values of $P_{ext}$, the pairwise correlations greatly increase the population Fano factor, which then decreases with increasing $a_{rec}$.

of the population activity:

$$\text{FF} = \frac{\text{Var}(\sum_i N_i)}{\text{E}[\sum_i N_i]} \tag{22}$$

$$= \frac{\sum_i \text{Var}(N_i)}{\sum_i \text{E}[N_i]} + \frac{2\sum_{i<j} \text{Cov}(N_i, N_j)}{\sum_i \text{E}[N_i]} \tag{23}$$

$$= \frac{\sum_i \text{Var}(N_i)}{\sum_i \text{E}[N_i]} \left(1 + \frac{2\sum_{i<j} \text{Cov}(N_i, N_j)}{\sum_i \text{Var}(N_i)}\right) \tag{24}$$

$$= \frac{\nu}{\mu}\left(1 + (n_{tot} - 1)\frac{c}{\nu}\right) \tag{25}$$

$$\approx \text{FF}_0(1 + kr) \tag{26}$$

where $c$ is the mean pairwise covariance, $\nu$ the mean variance of a neuron, $\mu$ is the mean number of spikes in $\Delta T$, $n_{tot}$ is the number of neurons, and $r$ is the Pearson correlation coefficient. The last approximation holds for neurons with identical variances and pairwise covariances [16]. It provides an insight into how the pairwise correlations and Fano factor of individual

neurons affect the Fano factor of the total activity. If the correlations or number of neurons are small ($r \cdot n_{\text{tot}} \ll 1$), the decorrelation by strengthening the recurrent synapses does not significantly decrease the population Fano factor. Instead, the population Fano factor may increase due to the increase of the Fano factor of individual neurons (Fig 2D, $P_{\text{ext}} = 0.01$). If greater correlations are induced due to the shared input to the network, the correlations have a dominating effect on the population Fano factor, which can then be greatly decreased by strengthening the recurrent synapses and in turn decreasing the pairwise correlations (Fig 2E and 2F).

## 2.4 Inhibitory feedback is metabolically costly

**2.4.1 Stronger recurrence strength increases the cost of the resting state.** We calculated the cost of the activity by summing the cost of action potentials from the excitatory, inhibitory, and external subpopulations, and the cost of excitatory synaptic currents in the excitatory and inhibitory subpopulations. These excitatory currents may be evoked by action potentials from the external or excitatory subpopulations, or from the background input. We did not consider the cost of synaptic currents evoked in neurons not involved in our simulation. We assume that such synaptic currents would be part of the background activity of a different area. Therefore, if we included these costs and considered multiple cortical areas, we would have included the background activity cost multiple times. We also did not include the cost of synaptic currents in the external population.

The cost of the resting state is an important factor for information-metabolic efficiency [10]. In our network, increasing the recurrence strength decreased the spontaneous activity of the neurons, due to inhibition dominating the recurrent currents. However, the simultaneous increase in the strength of the recurrent excitatory synapses increased the cost of the excitatory synaptic currents (Fig 3A, 3B and 3C), because the spontaneous action potentials from the excitatory subpopulation evoke stronger excitatory post-synaptic currents.

**2.4.2 Inhibitory feedback decreases gain.** Because the net current from recurrent synapses is hyperpolarizing, with stronger recurrent synapses, a stronger excitatory current is necessary to bring the neuron to a given post-synaptic firing rate, and higher pre-synaptic firing rates are necessary. Therefore, the gain $g$ of the network decreases, and with increasing $a_{\text{rec}}$ the cost of synaptic currents and the cost of external activity increase (Fig 3D and 3E).

## 2.5 Shared input decreases gain

The number of synapses from the external population for each neuron in the excitatory and inhibitory subpopulations follows a binomial distribution:

$$p(k) = \binom{n_{\text{ext}}}{k} P_{\text{ext}}^k (1 - P_{\text{ext}})^{n_{\text{ext}} - k}, \tag{27}$$

with the mean number of synapses given by $n_{\text{ext}} \cdot P_{\text{ext}}$ and variance $n_{\text{ext}} \cdot P_{\text{ext}}(1 - P_{\text{ext}})$. We scaled the firing rate of the individual neurons in the external population as $\lambda_{\text{exc}}^0 = \frac{\lambda_{\text{exc}}}{n_{\text{ext}} \cdot P_{\text{ext}}}$. Therefore the mean output to a single neuron was always $\lambda_{\text{ext}}$, independently of $P_{\text{ext}}$ and the variance of the input across neurons was $\lambda_{\text{ext}} n_{\text{ext}} \frac{1 - P_{\text{ext}}}{P_{\text{ext}}}$.

Given the convexity of the single neuron tuning curve in the analyzed input range (S1 Fig) that out of two inputs with an identical mean $\lambda_{\text{ext}}$, but different variances across neurons, the input with the higher variance will lead to a higher average firing rate. Assuming that the input across neurons follows a normal distribution with mean $\lambda_{\text{ext}}$ and variance $\sigma^2$ and that the single neuron tuning curve can be approximated by an exponential function in the form of $c_1 \exp$

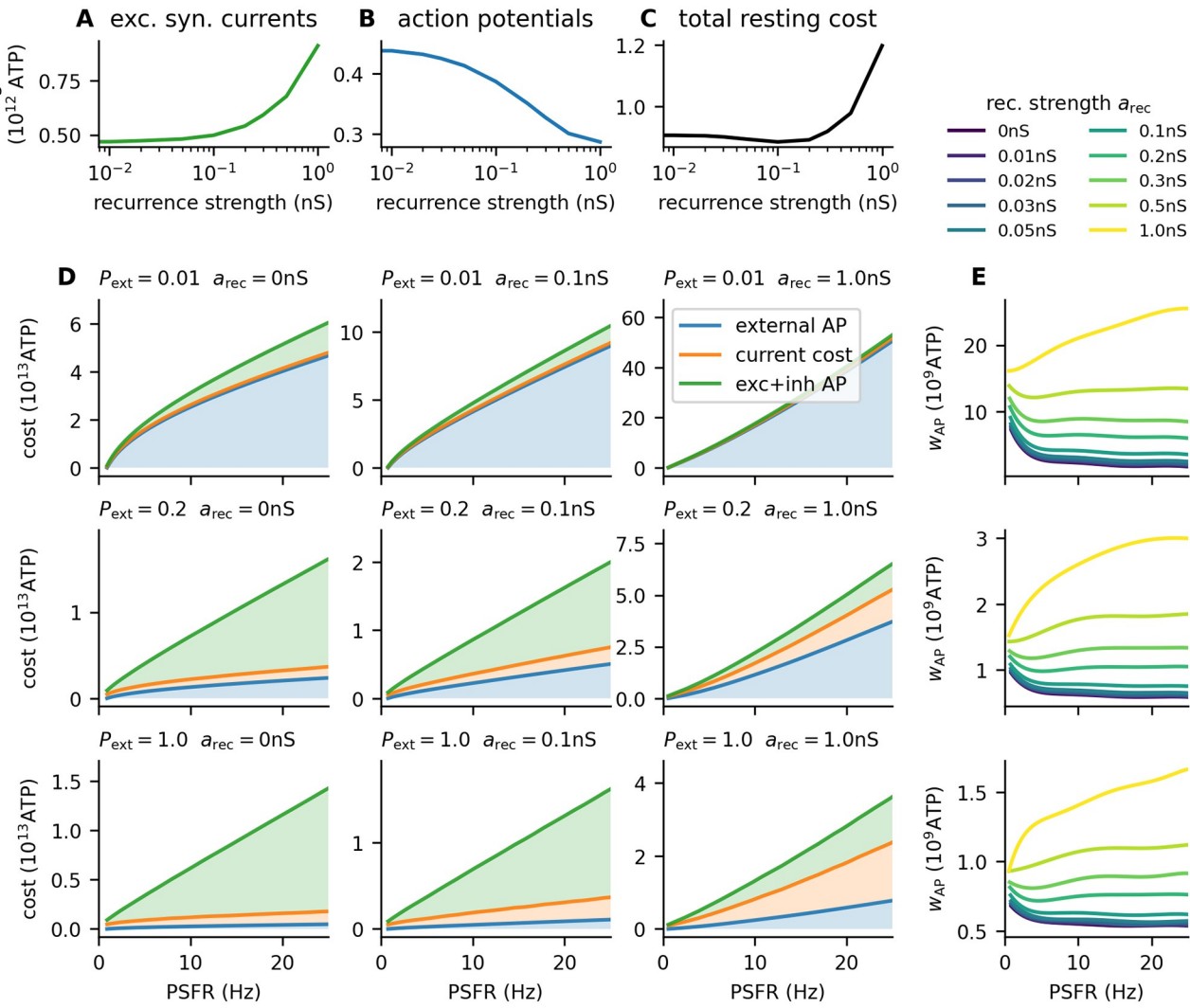

**Fig 3. Metabolic cost of the network activity. A-C**: Cost at resting state ($\lambda_{ext} = 0$). **A**: Cost of the excitatory synaptic currents from the background input (Eq 35) and excitatory action potentials evoked by the background input. **B**: Cost of the action potentials (both excitatory and inhibitory) evoked by the background input. **C**: Total resting cost obtained by summing **A** and **B**. **D**: The total cost of the network activity is plotted against the output of the network (the total post-synaptic firing rate). Filled areas represent individual contributions of each cost component: cost of action potentials from the external population, cost of the excitatory synaptic currents, and cost of the post-synaptic (evoked) action potentials. As $P_{ext}$ increases, the contribution of external action potentials to the overall cost decreases. With increasing $a_{rec}$, the contribution of excitatory synaptic currents increases. **E**: The cost of increasing the mean input by one action potential ($w_{AP}$, Eq 19) is significantly lower for higher $P_{ext}$. However, although the difference between $P_{ext} = 0.01$ and $P_{ext} = 0.2$ is approximately 10-fold, the difference between $P_{ext} = 0.2$ and $P_{ext} = 1$ is only approximately 2-fold, as the cost of the external population starts to contribute less to the overall cost.

($c_2x$), where $x$ is the input intensity to the single neuron, we obtain the mean firing rate:

$$\int_{-\infty}^{+\infty} \frac{1}{\sigma\sqrt{2\pi}} \exp\left[-\frac{(x-\lambda_{ext})^2}{2\sigma^2}\right] c_1 \exp(c_2 x) = \frac{c_1}{2} \exp\left(\frac{c_2}{2}\left(c_2\sigma^2 - 2\lambda_{ext}\right)\right), \tag{28}$$

which grows with the standard deviation of the input.

Accordingly, we observed that networks with higher $P_{ext}$ needed higher $\lambda_{ext}$ in order to produce the same mean PSFR as networks with lower $P_{ext}$ (Fig 4A, 4B and 4C), which translates to lower gain with higher $P_{ext}$ (Fig 4D, 4E and 4F). Moreover, the mean Fano factor of individual

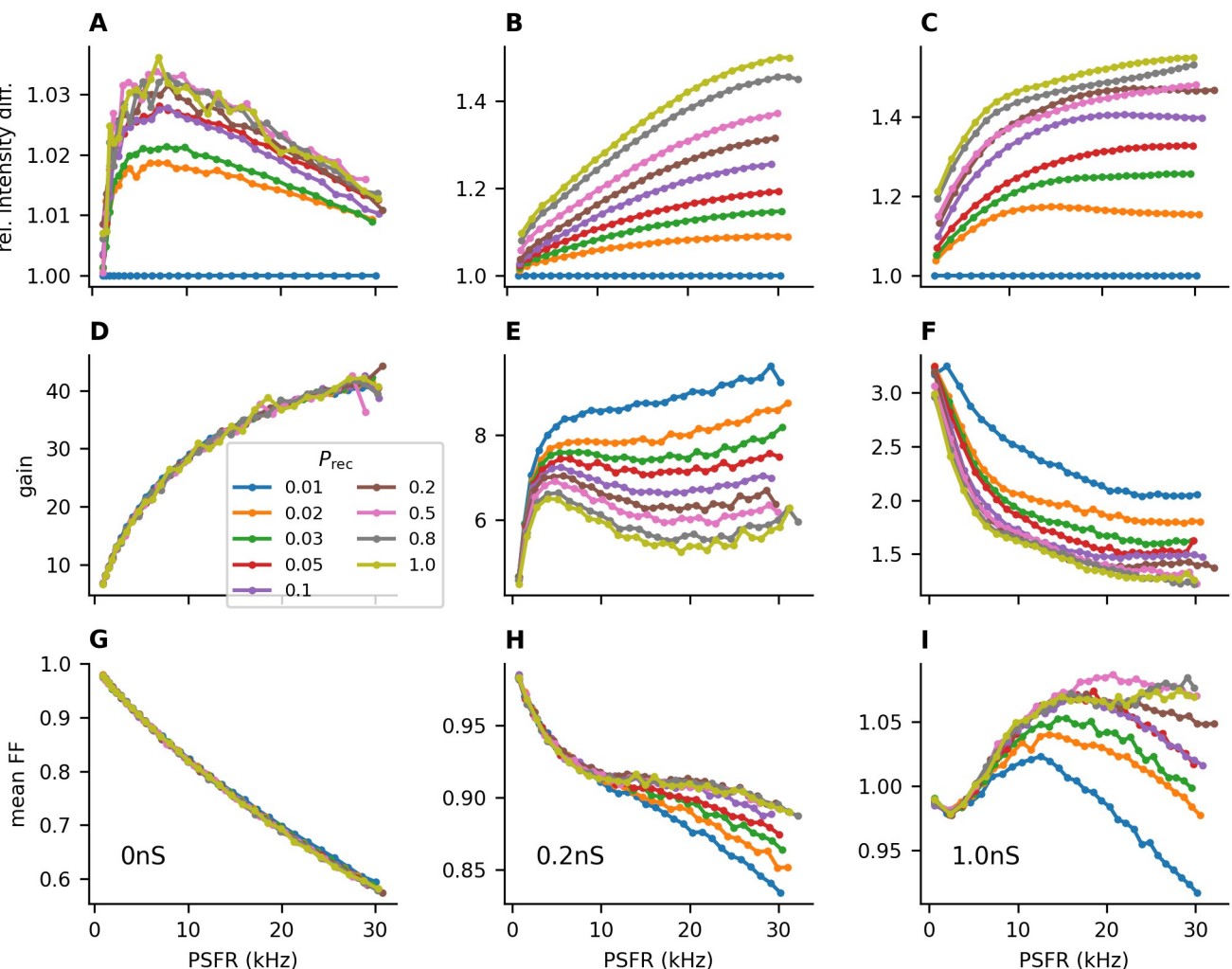

**Fig 4. Shared input decreases the gain and increases the individual Fano factor. A-C**: The input intensity $\lambda_{ext}$ needed to evoke a given firing rate (*x*-axis) with different connection probabilities $P_{ext}$ relative to the input intensity for $P_{ext} = 0.01$. **A**: $a_{rec} = 0$ nS, **B**: $a_{rec} = 0.2$ nS, **C**: $a_{rec} = 1$ nS. For higher $P_{ext}$, higher values of $\lambda_{ext}$ are needed to achieve the same post-synaptic firing rates as with lower values of $P_{ext}$. This effect becomes more pronounced in stronger recurrent synapses (**E-F**). **D-F**: Gain of the network (Eq 20). A higher $P_{ext}$ leads to a lower gain of the population activity. **G-I**: Higher values of $P_{ext}$ also increase the Fano factor of individual neurons.

neurons increased with increasing $P_{ext}$ (Fig 4G, 4H and 4I). This effect could be mostly removed by fixing the number of connections from the external population to each neuron in the excitatory and inhibitory populations to $P_{ext}n_{ext}$ (S2 Fig).

## 2.6 Optimal regimes for metabolically efficient information transmission

We illustrated that the recurrence strength 1) increases the metabolic cost of the neural activity and 2) decreases the population Fano factor by decreasing the correlations between the neurons. Similarly, the increased probability of a synapse from an external population ($P_{ext}$) decreases the cost of the neural activity but increases the noise correlations. The increased noise correlations then result in higher Fano factor (Eq 26). To find the balance between the cost of the network activity (Eq 4) and the mutual information between the input and the

output (Eq 5), we calculated the information-metabolic efficiency, which maximizes the ratio of the mutual information to the cost of the network activity (Eq 10).

For low values of $P_{\text{ext}}$ ($\leq 0.1$), increasing the strength of the recurrent input did not lead to an increase in the information-metabolic efficiency. For higher values of $P_{\text{ext}}$ the information-metabolic efficiency was maximized for $a_{\text{rec}}$ between 0.1 nS and 0.5 nS (Fig 5A and 5B), meaning that the strength of the recurrent excitatory synapses was 2× to 5× lower that the strength of the synapses from the external population.

Moreover, varying $P_{\text{ext}}$ had a significant effect on the information-metabolic efficiency across all values of $a_{\text{rec}}$. Namely, low values of $P_{\text{ext}}$ resulted in lower values of information-metabolic efficiency across all values of $a_{\text{rec}}$, showing that shared input from the external population is beneficial for metabolically efficient information transmission. Overall, the highest values of information-metabolic efficiency ($E \geq 2\text{bit}/10^{12}$ ATP) were reached for $a_{\text{rec}}$ between 0.05 nS and 0.5 nS and $P_{\text{ext}}$ between 0.2 and 1 (Fig 5B).

We analyzed the effect of the resting cost (Fig 3A, 3B and 3C) by setting the resting cost in all cases equal to $W_0$, the resting cost of the feedforward network. This did not have a significant effect on the information-metabolic efficiencies (S3 Fig).

Neural circuits might not necessarily maximize the ratio of information to cost. Instead, neurons and neural circuits could modulate their properties to maximize information transmission with the available energy resources [5]. For example, neurons in the mouse visual cortex have been shown to decrease the conductance of their synaptic channels after food restriction [35].

Accordingly, we studied how the optimal strength of recurrent synapses changes with the available resources. We calculated the optimal value of $a_{\text{rec}}$ for different values of available resources (3, 4, 5, 6, 7, 8, 10, 12, 15, 20, 30, and $40 \times 10^{12}$ ATP). In Fig 5C, 5D, 5E, 5F, 5G and 5H, we plotted $C(W; a_{\text{rec}})$, the capacity-cost function (Eq 9) extended by one dimension with $a_{\text{rec}}$. For each cost $W$, the optimal $a_{\text{rec}}$ is highlighted, and the corresponding contour of $C(W)$ is shown (see Table 1 for the values of $C(W)$). With decreasing $W$, the optimal value of $a_{\text{rec}}$ typically decreases. This effect is more robust with high values of $P_{\text{ext}}$, because the contours are more curved at the optimum.

We calculated the extended capacity-cost functions using input distributions obtained from the low-noise approximation. To verify that the low noise approximation applies in the case of the studied system, we compared these results to the information-metabolic efficiency obtained with the Jimbo-Kunisawa algorithm. The relative difference did not exceed 10% and did not have a significant impact on the information-metabolic efficiency heatmap structure (S4 Fig).

## 2.7 Limits of efficient information transmission by the population activity

So far we have assumed that the information about the stimulus is transmitted by the total activity of the network. Such analysis provides us with important insights, however, such simplistic decoding might not necessarily occur in the brain. To explore the limits of decoding the input intensity from the population activity, we assert that the brain can perform optimal unbiased decoding of the stimulus, i.e., for each stimulus $\lambda_{\text{ext}}$, it holds for the estimation of the input $\hat{\lambda}$ that

$$\text{E}[\hat{\lambda}] = \lambda_{\text{ext}}, \tag{29}$$

$$\text{Var}[\hat{\lambda}] = \frac{1}{J_{\text{pop}}(\lambda_{\text{ext}})}, \tag{30}$$

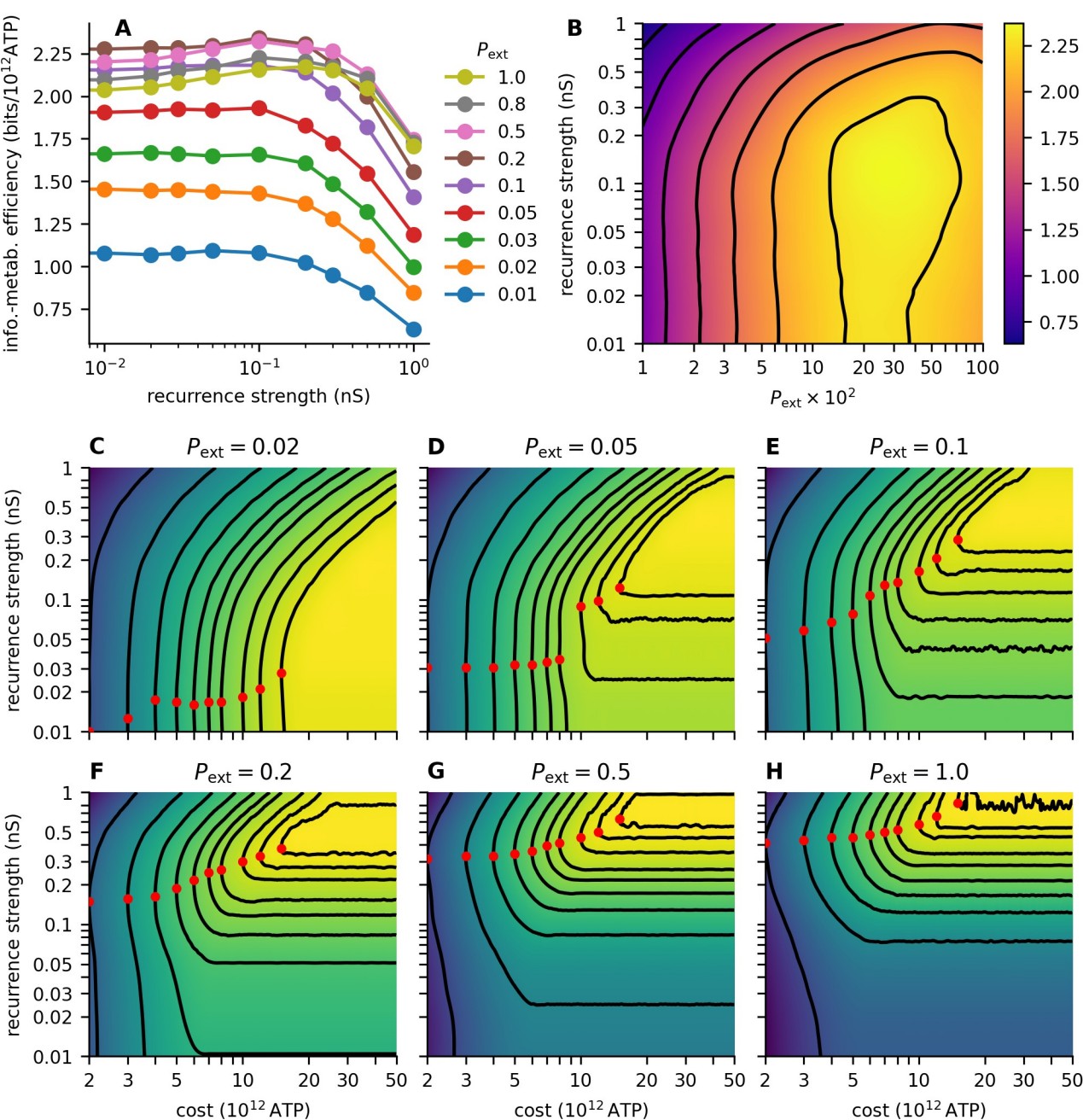

**Fig 5. Information transmission with cost constraints. A**: Information-metabolic efficiency $E$ (Eq 10) for different values of recurrence strength $a_{rec}$. $P_{ext}$ is color-coded. **B**: Contour plot of the information-metabolic efficiency. Contours are at 0.75, 1.0, 1.25, 1.5, 1.75, 2.0, and 2.25 bits/s. **C-H**: Contour plots showing the capacity-cost function $C(W)$ (Eq 9) with dependence on the recurrence strength $a_{rec}$ for different values of $P_{ext}$. The contours show the maximal capacities constraint at different values of $W$ (see Table 1 for the costs and capacity values at the contours). The heatmaps in **B-H** were calculated using piece-wise cubic 2D interpolation (SciPy interpolator CloughTocher2DInterpolator [34]) from the grid calculated with $P_{ext}$ values 0.01, 0.02, 0.03, 0.05, 0.1, 0.2, 0.5, 0.8, 1 and $a_{rec}$ values 0, 0.01, 0.02, 0.03, 0.05, 0.1, 0.2, 0.3, 0.5, and 1 nS.

**Table 1. Capacity-cost function values (in bits).**

| $P_{ext} \setminus W(10^{12}$ ATP) | 2 | 3 | 4 | 5 | 6 | 7 | 8 | 10 | 12 | 15 |
|---|---|---|---|---|---|---|---|---|---|---|
| 0.02 | 2.71 | 3.55 | 4.06 | 4.43 | 4.71 | 4.93 | 5.12 | 5.40 | 5.61 | 5.83 |
| 0.05 | 3.50 | 4.26 | 4.69 | 4.99 | 5.20 | 5.36 | 5.48 | 5.66 | 5.79 | 5.89 |
| 0.10 | 3.83 | 4.49 | 4.85 | 5.09 | 5.27 | 5.42 | 5.53 | 5.68 | 5.77 | 5.87 |
| 0.20 | 3.97 | 4.54 | 4.87 | 5.10 | 5.27 | 5.41 | 5.51 | 5.67 | 5.78 | 5.86 |
| 0.50 | 3.90 | 4.45 | 4.78 | 5.01 | 5.18 | 5.31 | 5.41 | 5.56 | 5.64 | 5.67 |
| 1.00 | 3.78 | 4.32 | 4.64 | 4.87 | 5.03 | 5.16 | 5.27 | 5.40 | 5.46 | 5.50 |

where the second equation corresponds to an estimator which saturates the Cramér-Rao bound, and $J_{pop}(\lambda_{ext})$ is the Fisher information about the stimulus from the population activity. If we assume that $\hat{\lambda}$ is distributed normally, we may then write the conditional probability distribution function as:

$$f(\hat{\lambda}|\lambda_{ext}) = \sqrt{\frac{J_{pop}}{2}} \exp\left[-\frac{J_{pop}}{2}(\lambda_{ext} - \hat{\lambda})^2\right], \qquad (31)$$

obtaining a noisy identity channel with the noise given by the Cramér-Rao bound.

To reduce the effect of sampling bias, we estimated $J_{pop}$ from the first 500 principal components of the output and employed a bias correction (see section 4.4 for details). Increasing the strength of recurrent connections ($a_{rec}$) increased the information metabolic efficiency of the network (Fig 6). The increase was more pronounced with higher values of $P_{ext}$, and overall was the highest for $P_{ext} = 0.8$ and $P_{ext} = 1$. In this sense, the results remain qualitatively very similar to the information-metabolic efficiency calculated from the summed activity (Fig 5). Interestingly, however, our results indicate that when using information from the entire population, not only the summed activity, the noise correlations introduced by the shared input are less detrimental, and $P_{ext} = 1$ reaches the highest or close to highest values of the information-metabolic efficiency.

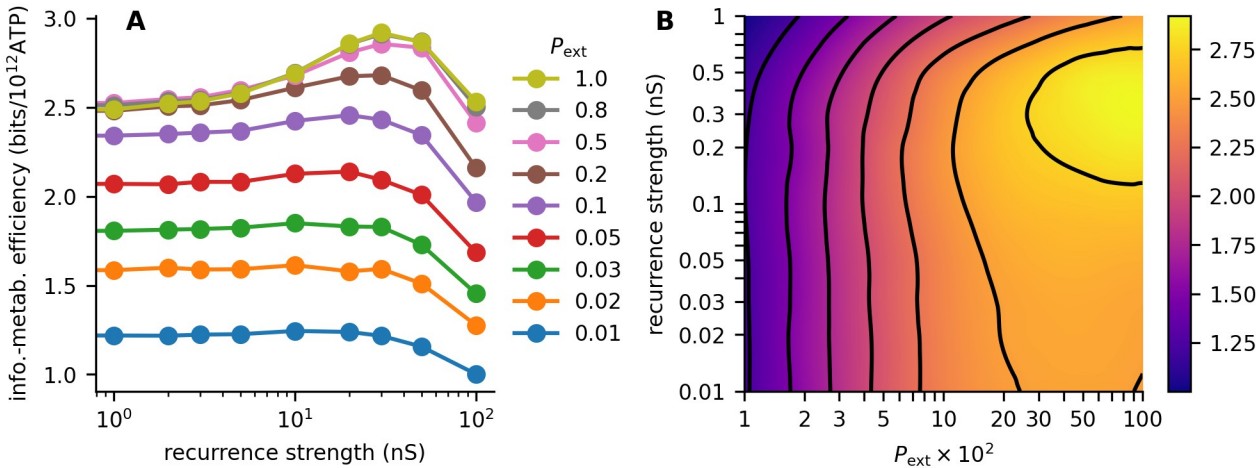

**Fig 6. Information-metabolic efficiency with multi-dimensional output. A**: Information-metabolic efficiency $E$ (Eq 10) for different values of recurrence strength $a_{rec}$. $P_{ext}$ is color-coded. **B**: Contour plot of the information-metabolic efficiency. Contours are at 1, 1.25, 1.5, 1.75, 2, 2.25, 2.5, and 2.75 bits/s.

## 3 Discussion

Information in the brain is likely transmitted by neuronal populations instead of single neurons [19]. One of the benefits is that by considering the signal from many neurons, it is possible to decrease the noise inherent to rate coding spiking neurons, and thus increase the information carried by the system. The information increase is however influenced by correlations between the neurons and their structure. In this work, we investigated a situation where a population of neurons tuned to the same stimulus transmits information about the stimulus intensity. In this case, positive noise correlations decrease the information carried by the population.

We parameterized the shared input with the probability of connection from the external population $P_{ext}$. Higher $P_{ext}$ means that the firing rate of neurons in the external population can be lower to maintain the same mean input to the information-transmitting population. This way, the shared input, while increasing the noise correlations, decreases the metabolic cost of the activity. In the studied system, we could mitigate the noise correlations by strengthening the recurrent connections and thus increasing the inhibitory feedback. However, to excite a population with inhibitory feedback requires stronger input than to excite a population without inhibitory feedback, and therefore, strengthening the recurrent connection increased the cost of the activity.

In our work, we studied the balance between increasing the transmitted information by decreasing the noise correlations and the associated increase in the cost of the activity. We showed that in a linear system, if the Fano factor of the population activity and the ratio $\frac{g}{w_0 \text{FF}}$ ($g$ is the gain of the system, or slope of the stimulus-response curve, $w_0$ is the slope of the stimulus-cost curve) remain constant, the cost-constrained capacity will remain constant as well.

We proceeded to calculate the stimulus-response relationship and the metabolic cost for a more biologically realistic neural system. In the studied system, the population Fano factor could not be considered constant. Instead, correlations between neurons increased with the mean output of the system, and the mean Fano factor of single neurons was also dependent on the mean output of the system, leading to complex dependence of the population Fano factor on the mean output of the system (Fig 2D, 2E and 2F). We found that despite increasing the noise correlations, the shared input helps with information-metabolically efficient information transmission. This was further accented if the noise correlations are decreased by the increase in the inhibitory feedback. Increasing the recurrence strength could lead to a 10% to 15% increase in the information-metabolic efficiency. The magnitude of the increase was dependent on the cost of the action potentials. If the cost of synaptic currents is negligible compared to the cost of the action potentials, there would be a higher benefit in increasing the inhibitory feedback since the increases in the cost of the synaptic current could also be neglected.

We illustrated the effect of inhibition-dominated recurrence and shared input on the metabolic cost of neural activity. An increased strength of recurrence increased the cost of excitatory synaptic currents due to the stronger excitatory synapses and stronger input from the external population, as well as the cost of the activity of the external population. A higher connection probability from the external population (higher shared input probability) led to a decrease in the external population activity cost, as the overall activity of the external population could be lower to result in the same mean input to the post-synaptic neurons. On the other hand, due to less variable input to single neurons with high values of $P_{ext}$, a higher mean input was required across all neurons to evoke the same mean post-synaptic activity.

In our model of the cortical area, we considered two neural subpopulations: excitatory and inhibitory. Each subpopulation was homogeneous, but we set the threshold of the inhibitory neurons lower to mimic the behavior of fast-spiking inhibitory neurons. The difference

between excitatory, regular spiking neurons and inhibitory, fast-spiking neurons is often described not only by differences in the threshold but also in differences in the adaptation properties [36, 37, 29]. In our case, we did not consider adaptation for simplicity because estimating the information capacity of a neural system with adaptation is computationally considerably more difficult [10].

In our work, we assumed that the neural circuit maximizes the mutual information between the input and the output neurons while minimizing the cost of the neural activity. Such an approach does not provide any information about how the information is encoded. It only calculates the limit on the amount of information that can be reliably transmitted. Yet, the principles of mutual information maximization have proven very useful in explaining the properties of neural systems. For example, the tuning curves of blowfly's contrast-sensitive neurons are adapted to the distribution of contrasts encountered in the natural environment [38]; the power spectrum of distribution of odor in pheromone plumes follows the power spectrum predicted for an optimal input to olfactory receptor neurons [39]; distributions of post-synaptic firing rates of single neurons during *in-vivo* recordings follow distributions predicted from cost-constrained mutual information maximization [40, 41, 42].

By assuming a particular coding scheme, it is possible to place further constraints on the complexity of information encoding, with the assumption that complex codes are not an efficient way to transmit information [43, 44]. We did not attempt this in our study. However, it would be interesting to study whether inhibitory feedback decreases or increases the encoding complexity.

We have shown that a cortical area can adapt to the amount of available energy resources. When resources are scarce, information transmission can be adapted by weakening the synaptic weights, thus expending fewer resources to reduce the noise correlations. Such a mechanism is implemented in the mouse visual cortex [35]. Padamsey et al. [35] showed that in food-restricted mice, the orientation tuning curves of individual orientation-sensitive neurons in the visual cortex become broader due to weakened synaptic conductances. In our work, we studied the properties of a neuronal population instead of single neurons. In particular, we considered a population encoding the stimulus intensity instead of the stimulus identity, such as orientation. An extension this model to a situation in which stimulus identity is encoded and shared input is introduced due to the overlap of receptive fields would be interesting.

Neurons recorded *in-vivo* typically exhibit a Fano factor close to 1.0 and constant over a broad range of post-synaptic firing rates [45, 46, 19]. In the optimal regimes with stronger recurrent synapses, the Fano factor decreased only very slowly over the studied range of post-synaptic firing rates (up to 30 Hz in a single neuron). With weaker synaptic strengths, the Fano factor of a single neuron decreases rapidly with an increasing post-synaptic firing rate. Our model predicts that fewer available resources would lead to weaker recurrent synapses. This hypothesis is straightforward to test by calculating the Fano factors during stimulus presentation (both population and single neuron) in food-restricted animals and comparing them to controls. We expect that the population Fano factor will increase (alternatively, the noise correlations will increase) with food scarcity, and single neuron Fano factors will decrease.

## 4 Methods

### 4.1 Network model

We modeled a network consisting of three subpopulations: external (ext), excitatory (exc), and inhibitory (inh). The external subpopulation consisted of Poisson neurons, defined by their firing intensity $\lambda_{ext}^0$ (same for all the neurons in the subpopulation). Neurons in the excitatory

and inhibitory subpopulations were modeled as leaky integrate-and-fire (LIF) neurons:

$$C_m \frac{\mathrm{d}V^i}{\mathrm{d}t} = g_L(E_L - V^i) + I_{\mathrm{rec}}^i(V^i, t) + I_{\mathrm{ext}}^i(V^i, t) + I_{\mathrm{bcg}}^i(V^i, t), \tag{32}$$

$$I_{\mathrm{rec}}^i(V^i, t) = g_{\mathrm{exc}}^i(E_e - V^i) + g_{\mathrm{inh}}^i(E_i - V^i), \tag{33}$$

$$I_{\mathrm{ext}}^i(V^i, t) = g_{\mathrm{ext}}^i(E_e - V^i), \tag{34}$$

$$I_{\mathrm{bcg}}^i(V^i, t) = g_{\mathrm{bcg,exc}}^i(E_e - V^i) + g_{\mathrm{bcg,inh}}^i(E_i - V^i), \tag{35}$$

$$\tau_{\mathrm{exc}} \frac{\mathrm{d}g_{\mathrm{ext}}^i}{\mathrm{d}t} = -g_{\mathrm{ext}}^i + \sum_{j=1}^{n_{\mathrm{ext}}} \sum_{t_s \in \mathcal{T}_{\mathrm{ext}}^j} W_{\mathrm{ext}}^{ij} \delta(t - t_s), \tag{36}$$

$$\tau_{\mathrm{exc}} \frac{\mathrm{d}g_{\mathrm{exc}}^i}{\mathrm{d}t} = -g_{\mathrm{exc}}^i + \sum_{j=1}^{n_{\mathrm{exc}}} \sum_{t_s \in \mathcal{T}_{\mathrm{exc}}^j} W_{\mathrm{exc}}^{ij} \delta(t - t_s), \tag{37}$$

$$\tau_{\mathrm{inh}} \frac{\mathrm{d}g_{\mathrm{inh}}^i}{\mathrm{d}t} = -g_{\mathrm{inh}}^i + \sum_{j=1}^{n_{\mathrm{inh}}} \sum_{t_s \in \mathcal{T}_{\mathrm{inh}}^j} W_{\mathrm{inh}}^{ij} \delta(t - t_s), \tag{38}$$

$$\tau_{\mathrm{exc}} \frac{\mathrm{d}g_{\mathrm{bcg,exc}}^i}{\mathrm{d}t} = (\mu_{\mathrm{bcg,exc}} - g_{\mathrm{bcg,exc}}^i) + \tau_{\mathrm{exc}}\sigma_{\mathrm{bcg,exc}} \sqrt{\frac{2}{\tau_{\mathrm{exc}}}} \eta_{\mathrm{exc}}^i(t), \tag{39}$$

$$\tau_{\mathrm{inh}} \frac{\mathrm{d}g_{\mathrm{bcg,inh}}^i}{\mathrm{d}t} = (\mu_{\mathrm{bcg,inh}} - g_{\mathrm{bcg,inh}}^i) + \tau_{\mathrm{inh}}\sigma_{\mathrm{bcg,inh}} \sqrt{\frac{2}{\tau_{\mathrm{inh}}}} \eta_{\mathrm{inh}}^i(t). \tag{40}$$

$I_{\mathrm{rec}}$ is the synaptic current arising from the recurrent connections (exc. to exc., exc. to inh., inh. to exc., inh. to inh.). $I_{\mathrm{ext}}$ is the excitatory current from external neurons. $I_{\mathrm{bcg}}$ is the current from synapses from neighboring cortex areas. $\mathcal{T}_{\mathrm{ext}}^j$, $\mathcal{T}_{\mathrm{exc}}^j$, $\mathcal{T}_{\mathrm{inh}}^j$ represent the spike times of the $j$-th external, excitatory, and inhibitory neuron respectively. The matrices $\mathbf{W}_{\mathrm{ext}}$, $\mathbf{W}_{\mathrm{exc}}$, $\mathbf{W}_{\mathrm{inh}}$ contain the synaptic connection strengths, $W_X^{ij} = a_X$ ($X \in \{\mathrm{ext, exc, inh}\}$) if the $j$-th neuron connects to the $i$-th neuron and 0 otherwise. The background (bcg) input from neighboring cortical areas is modeled as the Ornstein-Uhlenbeck process with means $\mu_{\mathrm{bcg,exc}}$ and $\mu_{\mathrm{bcg,inh}}$ and standard deviations of the limiting distributions $\sigma_{\mathrm{bcg,exc}}$ and $\sigma_{\mathrm{bcg,inh}}$ [47, 48]. We set the values of the background activity to match the moments of an exponential Poisson shot noise with rates $\lambda_{\mathrm{bcg,exc}} = 0.5$ kHz and $\lambda_{\mathrm{bcg,inh}} = 0.125$ kHz [49]:

$$\mu_X = a_X \tau_X \lambda_X, \tag{41}$$

$$\sigma_X = a_X \sqrt{\frac{\lambda_X \tau_X}{2}}, \tag{42}$$

where $X$ represents the excitatory or inhibitory background activity, leading to the ratio of inhibitory to excitatory conductance of $\frac{g\lambda_{\mathrm{bcg,inh}}}{\lambda_{\mathrm{bcg,exc}}} = 5$, as observed *in-vivo* [48] and a spontaneous firing rate of about 0.5 Hz to 1 Hz.

When the membrane potential $V$ crosses the firing threshold ($\theta_{\text{exc}}$, $\theta_{\text{inh}}$) a spike is fired and the membrane potential is reset to $E_L$.

The network consisted of $n_{\text{ext}} = 1000$ neurons in the external population, $n_{\text{exc}} = 800$ neurons in the excitatory population, and $n_{\text{inh}} = 200$ neurons in the inhibitory population. The connections were set randomly with connection probability for the recurrent connections (exc. to exc., exc. to inh., inh. to inh., inh. to exc.) set to $P_{\text{rec}} = 0.2$ and the connection probability from the external population (ext. to exc. and ext. to inh., $P_{\text{ext}}$) was varied from to 0.01 to 1 (Fig 1A). We created the connection matrices $\mathbf{W}_X$ by generating a matrix of random uniformly distributed numbers $\mathbf{R}_X$ from the interval [0, 1) and set $W_X^{ij} = a_X$ if $R_{\text{ext}}^{ij} < P_{\text{ext}}$ or $R_X^{ij} < P_X$ for $X \in \{\text{exc, inh}\}$. The random matrix $\mathbf{R}_{\text{ext}}$ was the same for all values of $P_{\text{ext}}$. In simulations where we controlled for the effects caused by a random number of connections from the external population, we fixed the number of connections by setting only the $k = n_{\text{ext}} P_{\text{ext}}$ elements in each row of $W_{\text{ext}}$ non-zero, in the location of the $k$ largest elements of the $i$-th row of $\mathbf{R}_{\text{ext}}$.

The simulations were carried out using the Brian 2 package [50] in Python with a 0.1 ms time step. Used parameters are given in Table 2.

## 4.2 Obtaining the input-output relationship of the network

We considered the total number of action potentials $n$ from the excitatory and inhibitory subpopulations in time window $\Delta T = 1$ s as the output of the network. We modeled the stimulus as the input from the thalamic neurons, parametrized by the mean input rate to a single neuron:

$$\lambda_{\text{ext}} = n_{\text{ext}} \lambda_{\text{ext}}^0 \frac{1}{P_{\text{ext}}}, \tag{43}$$

where $\lambda_{\text{ext}}^0$ is the firing rate of a single neuron in the external population, $n_{\text{ext}} \lambda_{\text{ext}}^0$ is the input firing rate at $P_{\text{ext}} = 1$, and $\frac{1}{P_{\text{ext}}}$ is a scaling factor to keep the mean input same regardless of $P_{\text{ext}}$. For each set of parameters ($a_{\text{rec}}$ and $P_{\text{ext}}$ pair) we determined the input $\lambda_{\text{ext}}^{\text{max}}(a_{\text{rec}}, P_{\text{ext}})$ for which the output reached 30 kHz. In order to obtain the input-output relationship, we discretized the input space into 30 equidistant stimulus intensities: $\lambda_{\text{ext}}^i(a_{\text{rec}}, P_{\text{ext}}) = \frac{i}{30} \lambda_{\text{ext}}^{\text{max}}(a_{\text{rec}}, P_{\text{ext}})$,

**Table 2. Parameters of the LIF model.**

| | | |
|---|---|---|
| Membrane capacitance | $C_m$ | 150 pF |
| Leak conductance | $g_L$ | 10 nS |
| Resting potential | $E_L$ | −80 mV |
| Exc. reversal potential | $E_e$ | 0 mV |
| Inh. reversal potential | $E_i$ | −80 mV |
| Exc. synapse decay | $\tau_{\text{exc}}$ | 5 ms |
| Inh. synapse decay | $\tau_{\text{inh}}$ | 5 ms |
| Exc. threshold | $\theta_{\text{exc}}$ | −55 mV |
| Inh. threshold | $\theta_{\text{inh}}$ | −60 mV |
| Ext. synapse amplitude | $a_{\text{ext}}$ | 1 nS |
| Exc. synapse amplitude | $a_{\text{exc}}$ | 0.01–1 nS |
| Inh. synapse amplitude | $a_{\text{inh}}$ | $g \cdot a_{\text{exc}}$ |
| Exc. inh. synapse amplitude | $a_{\text{bcg,exc}}$ | $a_{\text{ext}}$ |
| Bcg. inh. synapse amplitude | $a_{\text{bcg,inh}}$ | $g \cdot a_{\text{ext}}$ |
| Inh. scaling factor | $\alpha$ | 20 |

where $i = 0, \ldots, 30$. With a fixed network connectivity, we simulated the network 10800 times for each $\lambda_{\text{ext}}^i(a_{\text{rec}}, P_{\text{ext}})$.

We discretized the input space to 1000 equidistant stimulus intensities and estimated the mean output $\mu(\lambda_{\text{ext}})$ and variance $\sigma^2(\lambda_{\text{ext}})$ for each intensity by linear interpolation from the simulated data. We then estimated the input-output relationship, defined by the conditional probability distribution $f(n|\lambda_{\text{ext}})$ as a discretized normal distribution for each $\lambda_{\text{ext}}$, with corresponding mean and variance:

$$f(n|\lambda_{\text{ext}}) = \frac{1}{Z} \exp\left(-\frac{(x - \mu(\lambda_{\text{ext}})^2)}{\sigma^2}\right), \tag{44}$$

$$Z = \sum_{n=0}^{+\infty} \exp\left(-\frac{(x - \mu(\lambda_{\text{ext}})^2)}{\sigma^2}\right). \tag{45}$$

## 4.3 Metabolic cost of neural activity

In our calculations, we focus on the energy in the form of ATP molecules required to pump out Na$^+$ ions. We take into account the Na$^+$ influx due to excitatory post-synaptic currents, Na$^+$ influx during action potentials, and Na$^+$ influx to maintain the resting potential. To this end, we follow the calculations in [2] and [3], which we modify for our neuronal model.

We assume the standard membrane capacitance per area as $c_m = 1$ μF/cm$^2$ and the cell diameter as $D = 69$ μm, giving the total capacitance $C_m = \pi D^2 c_m = 150$ pF. Therefore, to depolarize a neuron by $\Delta V = 100$ mV the minimum charge influx is $\Delta V C_m = 1.5 \times 10^{-11}$ C and the minimum number of Na$^+$ ions $\frac{\Delta V C_m}{e} \doteq 9.375 \times 10^7$, where $e \doteq 1.6 \times 10^{-19}$ is the elementary charge. The minimal number of Na$^+$ ions is then quadrupled to get a more realistic estimate of the Na$^+$ influx due to the simultaneous opening of the K$^+$ channels [2]. The Na$^+$ influx must be then pumped out by the Na$^+$/K$^+$-ATPase, which requires one ATP molecule per 3 Na$^+$ ions. The cost of a single action potential can be then estimated as $\frac{4}{3} \times 9.375 \times 10^7$ ATP $= 1.25 \times 10^8$ ATP. However, about 75% of the metabolic costs associated with an action potential are expected to come from the propagation of the action potential through the neuron's axons [51, 2]. Therefore, we estimate the total cost as $5.0 \times 10^8$ ATP.

Next, we assume that the excitatory synaptic current is mediated by the opening of Na$^+$ and K$^+$ channels with reversal potentials $E_{\text{Na}} = 90$ mV and $E_{\text{K}} = -105$ mV. For the excitatory synaptic current, the following must hold

$$(g_{\text{exc}} + g_{\text{ext}})(V - E_e) = g_{\text{Na}}(V - E_{\text{Na}}) + g_{\text{K}}(V - E_{\text{K}}), \tag{46}$$

$$g_{\text{Na}} + g_{\text{K}} = g_{\text{ext}} + g_{\text{exc}}. \tag{47}$$

Therefore:

$$I_{\text{Na}} = \frac{g_{\text{K}}(V - E_{\text{K}})}{(g_{\text{exc}} + g_{\text{ext}})(V - E_e)}. \tag{48}$$

The sodium entering with the sodium current $I_{\text{Na}}$ must be pumped out by the Na$^+$/K$^+$-ATPase and therefore we calculate the cost of the synaptic current as $\frac{1}{3e} I_{\text{Na}} \Delta T$ ATP, where $\Delta T$ is the time interval over which we are measuring the cost.

Each input to the network (parametrized by $\lambda_{\text{ext}}$) is then associated with a cost, which we express as

$$w(\lambda_{\text{ext}}) \quad = \left( (N_{\text{exc}}\mu_{\text{exc}} + n_{\text{inh}}\mu_{\text{inh}} + n_{\text{ext}}\lambda_{\text{ext}}\frac{1}{P_{\text{ext}}})W_{\text{AP}} + + \frac{N_{\text{exc}}\langle I_{\text{Na}}^{\text{exc}}\rangle + n_{\text{inh}}\langle I_{\text{Na}}^{\text{inh}}\rangle}{3e} \right)\Delta T, \quad (49)$$

where $\mu_{\text{exc}} = \mu_{\text{exc}}(\lambda_{\text{ext}})$, $\mu_{\text{inh}} = \mu_{\text{inh}}(\lambda_{\text{ext}})$ are the mean firing rates of a single excitatory and inhibitory neuron (given the input $\lambda_{\text{ext}}$), $\langle I_{\text{Na}}^{\text{exc}}\rangle = \langle I_{\text{Na}}^{\text{exc}}\rangle(\lambda_{\text{ext}})$ and $\langle I_{\text{Na}}^{\text{inh}}\rangle = \langle I_{\text{Na}}^{\text{inh}}\rangle(\lambda_{\text{ext}})$ are the average excitatory synaptic currents in a single excitatory and inhibitory neuron.

## 4.4 Fisher information with multidimensional output

When we consider that the output of the network is either the full vector of firing rates, or its low-dimensional projection, we can calculate the Fisher information as

$$J_{\text{pop}}(\lambda_{\text{ext}}) = f'(\lambda_{\text{ext}})^T\Sigma^{-1}f'(\lambda_{\text{ext}}) + \frac{1}{2}\text{Tr}\left( \Sigma^{-1}\frac{\partial\Sigma^{-1}}{\partial\lambda_{\text{ext}}}\Sigma^{-1}\frac{\partial\Sigma^{-1}}{\partial\lambda_{\text{ext}}} \right), \quad (50)$$

where $\mathbf{f}(\lambda_{\text{ext}})$ is the mean of the multidimensional response vector, $\Sigma(\lambda_{\text{ext}})$ (dependence of $\Sigma$ was omitted for legibility) is the covariance matrix of the response components at input $\lambda_{\text{ext}}$, and Tr stands for the Trace operator. The first term in the equation is analogous to the Fisher information in one-dimensional case (Eq 14), while the second term indicates how much information we gain about the stimulus from changes in the covariance matrix. In our case, the second term was always very small compared to the first term.

We performed dimensionality reduction of the output across all stimuli by principal component analysis and used the first 500 principal components. We used 500, because the increase in information-metabolic efficiency for higher number of components is small, and the sampling bias is still relatively small (S5 Fig). To deal with the remaining sampling bias we calculated the information-metabolic efficiency with the Jimbo-Kunisawa for different numbers of trials and performed the quadratic extrapolation method to estimate the unbiased information-metabolic efficiency [52, 53]. Overall, the results remain qualitatively very similar to the information-metabolic efficiency calculated from the summed activity. However, we found the increase in information-metabolic efficiency from using high-dimensional output is the largest for higher values $a_{\text{rec}}$ and $P_{\text{ext}}$.

**4.4.1 Correcting the sampling bias.** In the case of a high-dimensional output, insufficient number of trials may lead to perceived correlations in the data which are in fact not there, subsequently increasing the calculated mutual information [54, 52, 55, 56, 53]. To decrease the sampling bias, we first performed principal component analysis to decrease dimensionality of the output and employed a quadratic extrapolation method to estimate the unbiased value of information-metabolic efficiency. We used the Jimbo-Kunisawa algorithm to calculate information-metabolic efficiency with 10800, 5400, and 2700 trials, obtaining the estimates of $E$ (Eq 10): $E_{10800}$, $E_{5400}$, and $E_{2700}$. We then assumed that the estimates follow the following dependency on the number of trials $k$ [52]:

$$E_k = E_0 + \frac{a}{k} + \frac{b}{k^2}. \quad (51)$$

By solving the linear system we obtained the estimate of the unbiased information-metabolic efficiency $E_0$ (S5 Fig). We found that with 500 principal components the bias is still relatively low, and further increasing the number of components leads only to minor increase in the information-metabolic efficiency. Therefore, we used the first 500 components to obtain the results in the Fig 6.

## Supporting information

**S1 Fig. Input-output relationship of a single neurons.** To exclude the network effects, we plotted the tuning curves for the feedforward network separately for the excitatory (blue) and inhibitory (yellow) neurons. The thick line represents the median response across the neurons, which shows that their tuning curves are convex in the studied range. The shaded area shows the spread of the tuning curves across neurons (2.5 to 97.5 percentile). With low values of $P_{ext}$, the tuning curves across neurons vary significantly and are skewed to the higher firing rates. (TIF)

**S2 Fig. Fixing the number of external connections to each neuron.** Same as Fig 4, but exactly $k_{ext}P_{ext}$ external neurons connected to each excitatory and inhibitory neuron. This removed a large part of the dependence on $P_{ext}$ seen in Fig 4. (TIF)

**S3 Fig. Effect of equalizing the resting cost on the information-metabolic efficiency.** We observed that the cost of the resting state was different for different recurrence strengths $a_{rec}$ (Fig 3A–3C). This could potentially explain the higher information-metabolic efficiency $E$ (Eq 10) for intermediate values of $a_{rec}$ and its decrease for high values of $a_{rec}$. To quantify the effect of the resting cost, we set the resting cost in each case to the resting cost of the feedforward network $W_0(a_{rec} = 0)$. The differences in the cost of the resting state did not have a qualitative effect on the conclusions. **A**: The same contour plot as in Fig 5B. **B**: Contour plot with equalized resting costs (contours as in Fig 5B: 0.75, 1.0, 1.25, 1.5, 1.75, 2.0, and 2.25 bits/s). **C**: Heatmap of the relative differences. (TIF)

**S4 Fig. Accuracy of information-metabolic efficiency approximation.** To calculate the capacity-cost functions, we calculated the mutual information using Eq (5) with the input probability distribution calculated from Eqs (12) and (14). Here we compare the information-metabolic efficiencies calculated with the approximation and the Jimbo-Kunisawa algorithm. **A**: The same contour plot as in Fig 5B with information-metabolic efficiencies calculated with the Jimbo-Kunisawa algorithm. **B**: Information-metabolic efficiencies calculated with the Fisher-information-based input distribution. **C**: Heatmap of the relative differences. Note that the approximation can only reach values lower than the actual information-metabolic efficiency. (TIF)

**S5 Fig. Sampling bias and extrapolation.** The information-metabolic efficiency calculated by the Jimbo-Kunisawa algorithm is plotted for different numbers of principal components used. We calculated the information-metabolic efficiency from different numbers of trials. At high number of components, lower number of trials lead to significantly higher information-metabolic efficiency. This is the effect of the sampling bias. We attempted to remove the bias by using the quadratic extrapolation method. For 500 principal components the bias is still relatively low, and increasing the number of components brings little benefit in terms of information-metabolic efficiency. (TIF)

## Author Contributions

**Conceptualization:** Tomas Barta, Lubomir Kostal.

**Formal analysis:** Tomas Barta.

**Investigation:** Tomas Barta.

**Software:** Tomas Barta.

**Supervision:** Lubomir Kostal.

**Writing – original draft:** Tomas Barta.

**Writing – review & editing:** Tomas Barta, Lubomir Kostal.

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
