## [Decision Letter · Decision Letter 0]

13 Jun 2023

Dear Mr. Barta,

Thank you very much for submitting your manuscript "Shared input and recurrency in neural networks for metabolically efficient information transmission" for consideration at PLOS Computational Biology.

As with all papers reviewed by the journal, your manuscript was reviewed by members of the editorial board and by several independent reviewers. In light of the reviews (below this email), we would like to invite the resubmission of a significantly-revised version that takes into account the reviewers' comments.

Both reviewers find the topic interesting and value the approach. However, both reviewers also raise some concerns with the derivations and with the writing, that both need some clarification. Also, reviewer 1 offers some suggestions to extend the scope of this manuscript. I believe these issues can be dealt with by the authors.

We cannot make any decision about publication until we have seen the revised manuscript and your response to the reviewers' comments. Your revised manuscript is also likely to be sent to reviewers for further evaluation.

Sincerely,

Fleur Zeldenrust

Academic Editor

PLOS Computational Biology

Daniele Marinazzo

Section Editor

PLOS Computational Biology

Both reviewers find the topic interesting and value the approach. However, both reviewers also raise some concerns with the derivations and with the writing, that both need some clarification. Also, reviewer 1 offers some suggestions to extend the scope of this manuscript. I believe these issues can be dealt with by the authors.

Reviewer's Responses to Questions

**Comments to the Authors:**

Reviewer #1: Please refer to the comments in the PDF.

Reviewer #2: The review has been uploaded as an attachment.

**Have the authors made all data and (if applicable) computational code underlying the findings in their manuscript fully available?**

Reviewer #1: Yes

Reviewer #2: Yes

PLOS authors have the option to publish the peer review history of their article (what does this mean?). If published, this will include your full peer review and any attached files.

Reviewer #1: No

Reviewer #2: No
---

## [Decision Letter · Decision Letter 1]

9 Jan 2024

Dear Barta,

Thank you very much for submitting your manuscript "Shared input and recurrency in neural networks for metabolically efficient information transmission" for consideration at PLOS Computational Biology. As with all papers reviewed by the journal, your manuscript was reviewed by members of the editorial board and by several independent reviewers. The reviewers appreciated the attention to an important topic. Based on the reviews, we are likely to accept this manuscript for publication, providing that you modify the manuscript according to the review recommendations.

Both reviewers agree that almost everything is dealt with. Only Reviewer 1 still has some minor issues, that I believe can be easily dealt with. Once these have been addressed, I believe the manuscript will be ready for publication.

Sincerely,

Fleur Zeldenrust

Academic Editor

PLOS Computational Biology

Daniele Marinazzo

Section Editor

PLOS Computational Biology

Both reviewers agree that almost everything is dealt with. Only Reviewer 1 still has some minor issues, that I believe can be easily dealt with. Once these have been addressed, I believe the manuscript will be ready for publication.

Reviewer's Responses to Questions

**Comments to the Authors:**

Reviewer #1: Please see attached PDF

Reviewer #2: I thank the authors for making the appropriate changes and I believe the paper reads much better now.

**Have the authors made all data and (if applicable) computational code underlying the findings in their manuscript fully available?**

Reviewer #1: Yes

Reviewer #2: Yes

PLOS authors have the option to publish the peer review history of their article (what does this mean?). If published, this will include your full peer review and any attached files.

Reviewer #1: **Yes: **Michele Nardin

Reviewer #2: No

Figure Files:

Data Requirements:

Reproducibility:

References:

---

## [Editor Report · Decision Letter 2]

7 Feb 2024

Dear Barta,

We are pleased to inform you that your manuscript 'Shared input and recurrency in neural networks for metabolically efficient information transmission' has been provisionally accepted for publication in PLOS Computational Biology.

Best regards,

Fleur Zeldenrust

Academic Editor

PLOS Computational Biology

Daniele Marinazzo

Section Editor

PLOS Computational Biology

---

## [Editor Report · Acceptance letter]

20 Feb 2024

PCOMPBIOL-D-23-00763R2 

Shared input and recurrency in neural networks for metabolically efficient information transmission

Dear Dr Barta,

I am pleased to inform you that your manuscript has been formally accepted for publication in PLOS Computational Biology. Your manuscript is now with our production department and you will be notified of the publication date in due course.

With kind regards,

Judit Kozma
